# Comparison of BSGI and MRI as Approaches to Evaluating Residual Tumor Status after Neoadjuvant Chemotherapy in Chinese Women with Breast Cancer

**DOI:** 10.3390/diagnostics11101846

**Published:** 2021-10-06

**Authors:** Hongbiao Liu, Hongwei Zhan, Ying Zhang, Gangqiang He, Hui Wang, Qiaoxia Zhang, Lili Zheng

**Affiliations:** Department of Nuclear Medicine, The Second Affiliated Hospital, Zhejiang University School of Medicine, Hangzhou 310009, China; zhangying5401@zju.edu.cn (Y.Z.); hegangqiang75@163.com (G.H.); qing26ganlan@163.com (H.W.); qiaoxiazhang688@163.com (Q.Z.); 2002ke@163.com (L.Z.)

**Keywords:** breast-specific gamma imaging (BSGI), neoadjuvant chemotherapy (NAC), breast cancer, magnetic resonance imaging (MRI), residual tumor size

## Abstract

Background: The present retrospective study was designed to evaluate the relative diagnostic utility of breast-specific gamma imaging (BSGI) and breast magnetic resonance imaging (MRI) as means of evaluating female breast cancer patients in China. Methods: A total of 229 malignant breast cancer patients underwent ultrasound, mammography, BSGI, and MRI between January 2015 and December 2018 for initial tumor staging. Of these patients, 73 were subsequently treated via definitive breast surgery following neoadjuvant chemotherapy (NAC), of whom 17 exhibited a complete pathologic response (pCR) to NAC. Results: BSGI and MRI were associated with 76.8% (43/56) and 83.9% (47/56) sensitivity (BSGI vs. MRI, *p* = 0.341) values, respectively, as a means of detecting residual tumors following NAC, while both these approaches exhibited comparable specificity in this diagnostic context. The specificity of BSGI for detecting residual tumors following NAC was 70.6% (12/17), and that of MRI was 58.8% (10/17) (BSGI vs. MRI, *p* = 0.473). Conclusion: These results demonstrate that BSGI is a useful auxiliary approach to evaluating pCR to NAC treatment.

## 1. Background

Breast cancer is the third most prevalent cancer type, affecting roughly 2 million people per year, including 1.9 million women in 2017 alone, and remaining the leading cancer-related cause of death in this demographic group [1]. While early-stage breast cancer can often be effectively treated via radical surgery, neoadjuvant chemotherapy (NAC) is often required as a means of controlling tumor progression prior to breast-conserving radical surgery in those with locally advanced disease. NAC can significantly increase rates of overall and disease-free survival (OS and DFS, respectively) similarly to postoperative chemotherapy, while increasing rates of breast-conserving surgery in those with operable locally advanced disease [2,3]. NAC treatment can also decrease the extent of resection in cases where tumors are over 2 cm in size. The amount of residual tumor remaining following NAC is an important prognostic indicator in treated patients [4]. Magnetic resonance imaging (MRI) is generally considered to be the ideal imaging modality when diagnosing, staging, and monitoring breast cancer in individuals being treated with NAC, and MRI-based analyses of post-NAC residual tumor have been reported to be more accurate than similar analyses made via mammography, ultrasound, or palpation [5,6,7]. However, non-mass enhancement at pre-treatment MRI negatively affected the diagnostic performance of MRI in assessing treatment response after NAC. MRI can also not be used regularly owing to its high costs [8,9]. In addition, patients with claustrophobia or magnetic resonance contraindications such as pacemaker installation cannot undergo magnetic resonance examination. Therefore, supplementary methods of MRI must be found. Breast-specific gamma imaging (BSGI) is a high-resolution radioimaging strategy that enables breast tissue visualization using a gamma camera with a limited field-of-view, which can be used to precisely detect breast cancer in tissues of variable density. Herein, we compared the relative diagnostic accuracy of MRI and BSGI as approaches to predicting complete pathologic response (pCR) and residual tumor size in patients undergoing NAC in an effort to provide clinical guidance for physicians.

## 2. Methods

### 2.1. General Information

The ethics committee of our hospital (Second Affiliated Hospital of Zhejiang University School of Medicine, Hangzhou, China) approved (2020-061; 22 January 2020) the present retrospective study. Data from 390 women who underwent BSGI at our hospital from January 2015–December 2018 were assessed. All these patients had been assessed via ultrasound, mammography, MRI, and BSGI prior to diagnosis in order to facilitate formal clinical staging. Among them, 229 patients were diagnosed with malignant tumors, of whom 73 were subsequently treated via definitive breast surgery following NAC treatment. Patient medical records were reviewed to extract key clinicopathological information, including age, tumor location, size, nuclear grade, and histological type. MRI and BSGI were conducted to detect residual tumors before and following NAC.

### 2.2. MRI

All MRI scans were conducted with patients in the prone position with a 1.5T system (Siemens, Erlangen, Germany) and a dedicated breast coil. Multiple contiguous axial and sagittal T1-weighted unenhanced and contrast-enhanced images (with and without fat suppression) and axial and sagittal images T2-weighted images were obtained. Reconstructed 3D maximum intensity and subtraction imaging were also performed. Residual tumors were defined based upon observed reductions in tumor enhancement and/or size when comparing MRI scans to those collected prior to NAC. Complete response (CR) was defined by total interval resolution of the previously detected lesion. MRI scans were evaluated by two radiologists based upon BI-RADS classification criteria, with any inconsistencies in their evaluations being resolved via discussion and consensus.

### 2.3. BSGI

Patients did not undergo any specific preparation for BSGI evaluation and maintained a normal diet. Patients were injected via an antecubital vein contralateral to the breast lesion with 555–740 MBq (15–20 mCi, Shanghai GMS Pharmaceutical Co., Ltd., Shanghai, China) ^99m^Tc-sestamibi. BSGI was then performed 10 min post-injection with patients in a seated position via the use of a breast-specific gamma camera (Dilon 6800; Dilon Technologies, Newport News, VA, USA). High-resolution bilateral craniocaudal (CC) and mediolateral oblique (MLO) images. Individual image acquisition was conducted for approximately 5 min, with a minimal range of 100 Kcounts/image [9,10]. Two nuclear medicine specialists evaluated BSGI results as per the Society of Nuclear Medicine guidelines [11] for interpreting BSGI results while also considering available baseline ultrasound and mammography images. The presence of residual viable tumor was first assessed via visual analysis, and any viable tumors were measured based upon the longest diameter in CC and MLO images, which was defined as the tumor size. When patients exhibited multifocal breast cancer, the diameter of the largest individual tumor was measured [12,13]. Inconsistencies were resolved through discussion and consensus. The baseline results of BSGI were determined according to the Society of Nuclear Medicine guidelines, and grade 4–5 was determined to be positive. The presence of residual tumor in BSGI images was defined by a location of a known previous tumor that exhibited a reduction in intensity or size relative to baseline but that exhibited mild or greater regional radiotracer uptake. CR was defined by an absence of any radiotracer uptake in a region known to correspond to the location of a prior tumor.

### 2.4. Pathological Assessment

Breast tumor pathological characteristics were defined as per the World Health Organization (WHO) classification system. Tissue samples that were resected following NAC treatment were subjected to hematoxylin and eosin (H&E) staining and were evaluated for evidence of residual invasive microscopic or macroscopic carcinoma, histologic nuclear grade, presence of lymphovascular invasion, and margin status. Immunohistochemical staining following diagnostic biopsy prior to NAC treatment was used to evaluate tumor hormone receptor and HER2 status. Tumors were separated into four subtypes based upon HER2, Ki-67, progesterone receptor (PR), and estrogen receptor (ER) status as follows: Luminal A, Luminal B, Her-2 positive, and triple-negative [9,14].

CR, partial response (PR), stable disease (SD), and progressive disease (PD) were defined as per the tumor size response evaluation criteria in solid tumors (RE-CIST) criteria. An absence of visible target lesions and a decrease in any target or non-target pathological lymph nodes to a short axis of <10 mm was used to define pCR [15]. Patients exhibiting partial responses or no response were categorized as ‘non-pCR’ for the purposes of the present study. The final measurement of the residual tumor prior to surgery was used for analytical purposes. Pathological evaluation following NAC treatment was conducted according to the Miller–Payne (MP) grading system, with five defined grades [16,17]: Grade 1, unchanged tumor cell density; Grade 2, <30% density reduction; Grade 3, 30–90% density reduction; Grade 4, >90% density reduction 90%; Grade 5, tumor cells were no longer visible.

### 2.5. Chemotherapy Regimens

NAC regimens were composed of epirubicin and cyclophosphamide (EC); docetaxel, epirubicin, and cyclophosphamide (TEC); docetaxel, carboplatin, and trastuzumab (TCH); docetaxel and trastuzumab (TH); epirubicin and cyclophosphamide (EC)/ docetaxel and trastuzumab (TH); doxorubicin; and cyclophosphamide and docetaxel (ACT). Patients included in the present study underwent 4–8 NAC treatment cycles. In total, 3, 8, 11, 2, 24, and 25 patients underwent TH, EC/TH, TCH, ACT, EC, and TEC regimen treatments, respectively.

### 2.6. Statistical Analysis

Sensitivity and specificity values were calculated for both MRI and BSGI. Pathologic examination measurement results served as a gold standard for the present study and were compared to the sizes of tumors as measured via BSGI and MRI, and the agreement between BSGI and MRI was measured by the Bland–Altman plots. Chi-squared tests were used to compare data. Quantitative data were given as means ± standard deviation (X ± s) when normally distributed. Data were analyzed using SPSS v22.0 (IBM Corp., Armonk, NY, USA), and *p* = 0.05 was the significance threshold.

## 3. Results

### 3.1. Patient Characteristics

In total, 390 women were evaluated via mammography, ultrasound, and BSGI at our hospital between January 2015 and December 2018, of whom 235 had also been evaluated via MRI. Of these patients, 229 were diagnosed with breast cancer. We identified 73 of these patients as being eligible for inclusion in the present retrospective study, as they had undergone both MRI and BSGI prior to NAC and had undergone definitive breast surgery following NAC. The average age of these patients was 52.8 years (range: 25–74) (Table 1). Core needle biopsy-confirmed axillary nodal metastases were detected in 18 patients (24.7%), while 17 (23.3%) exhibited pCR following NAC treatment. These 17 patients included 16.7% (2/12) of patients with Luminal A disease, 21.2% (7/33) patients with Luminal B disease, 31.8% (7/22) patients with HER2+ disease, and 16.7% (1/6) patients with triple-negative disease. Just 12 of these 17 patients exhibited no residual tumor or ductal carcinoma in situ (DCIS), while the remaining five exhibited no residual invasive carcinoma but did present with a small focus of residual DCIS. 

### 3.2. Residual Tumor Detection Following NAC

A athologically confirmed residual viable tumor was detected in 56 of these 73 patients (76.7%). BSGI and breast MRI scans exhibited respective sensitivity values of 76.8% (43/56) and 83.9% (47/56) for the detection of residual tumor, while corresponding specificity values were 70.6% (12/17) and 58.8% (10/17), respectively. Of the five cases that yielded false-positive results via BSGI imaging, four had HER2+ disease and Ki-67 >50%, while the remaining patient had Luminal A disease. Mild uptake was observed in three cases, with a tumor to normal tissue (T/N) ratio ranging from 1.2–1.4. Significant differences in the T/N value of the pCR group were observed when comparing values before and after NAC (*p* = 0.044) (Table 2). The maximum diameter of the tumor was 3.49 ± 1.63 cm before NAC by MRI exhibited, range from 1.03 to 9.67 cm, and it was 3.77 ± 1.73 cm (1.21–9.83 cm) by BSGI exhibited (*p* = 0.298). After NAC, the maximum diameter of the residual tumor was 1.92 ± 1.59 cm by MRI exhibited, range from 0.22 to 6.58 cm, and it was 1.90 ± 1.73 cm (0.59~8.33 cm) by BSGI exhibited (*p* = 0.900) (Table 3). Bland–Altman plots confirm the consistency of the two modalities (Figure 1).

BSGI sensitivity exhibited a significant correlation with residual tumor cellularity (Table 4) such that for tumors with cellularity > 10% the sensitivity was 94.3% (33/35), which differed significantly from that of tumors with cellularity ≤ 10% (*p* = 0.047). BSGI sensitivity was 79.2% (19/25) for tumors with a maximal residual diameter of ≤ 15mm, whereas it was 96.8% (30/31) for residual tumors > 15 mm in size (*p* = 0.019). All 13 false-negative patients with a residual tumor ≤ 15 mm exhibited low-residual tumor cellularity (seven patients with cellularity ≤10%, three with cellularity = 20–30%, three with cellularity = 40–60%). Of these false-negative patients, nine were detected via MRI. MRI scans exhibited similar sensitivity as a function of residual tumor cellularity (Table 4), with good statistical consistency between BSGI and MRI results.

### 3.3. False-CR, False-PR, and False-PD BSGI Findings

In total, 14, 6, and 2 false-CR, false-PR, and false-PD BSGI findings, respectively, were identified in the present patient cohort. In these patients, there were also 12 cases were incorrectly identified by MRI. There was one case with Liminal A and four cases with HER2 positive in the pCR group (Figure 2). There were five cases with Luminal B and two cases with HER2 positive in the non-pCR group. The evaluation of BSGI, MRI, and pathological measurements of residual tumor following NAC treatment were shown in Table 5.

## 4. Discussion

This study is the first we are aware of to have compared the relative diagnostic utility of MRI and BSGI when detecting residual tumors in female Chinese breast cancer patients following NAC treatment. We determined that BSGI represents an effective approach to the detection and evaluation of suspect lesions. NAC has been used with increasing frequency in recent years as a treatment for women presenting with locally advanced breast cancer or regional lymph node metastases. Such chemotherapeutic treatment can reduce tumor size and staging, improving breast-conserving surgery rates and facilitating the improved comprehensive treatment of early-stage breast cancer. However, not all patients respond to NAC. In addition, overestimates of tumor size may result in patients undergoing unnecessary surgery and suffering from an increased psychological burden, while underestimating tumor progression can result in delayed treatment. Therapeutic efficacy must therefore be evaluated as quickly and accurately as possible so that appropriate interventional strategies can be formulated while minimizing the risk of toxicity. Breast MRI scans are one of the primary approaches used to gauge patient responses to NAC; however, breast MRI analyses can nonetheless yield high false-positive rates and can overestimate tumor size, whereas BSGI is a more recently developed imaging approach that has been leveraged for both tumor staging and NAC response evaluation [9,13,16]. BSGI is a functional imaging approach that utilizes the positively-charged, fat-soluble ^99m^Tc-sestamibi radiotracer to estimate mitochondrial metabolic activity. After intravenous injection, this radiotracer can passively enter cells via transmembrane transport, whereupon it is attracted to the mitochondrial membrane owing to its positive charge. Mitochondrial numbers are closely correlated with cellular activity levels, such that tumor cells with a highly active metabolism take up more ^99m^Tc-sestamibi relative to normal cells [18]. The degree of radiotracer accumulation within breast tumors is reflective of tumor cell proliferation and malignancy, offering a more sensitive approach to gauging such activity as compared to morphological examination.

In previous reports, BSGI has been reported to have respective sensitivity and specificity values of 83–100% and 70.0–87.9% for diagnosing breast cancer [19,20], while for the detection of residual tumor following NAC treatment, these values range from 70–74.0% and 72.2–90%, respectively [13,14]. Herein, we found BSGI to exhibit respective sensitivity and specificity values of 75.0% and 70.6% for residual tumor detection, in line with prior reports and similar to the diagnostic performance of MRI in this context. The expression level of Ki-67 reflects the proliferative activity of cells, which is closely related to their chemosensitivity. Chen [21] and Ács [22] found that higher Ki-67 positivity prior to NAC treatment was predictive of pCR in breast cancer patients, but such elevated Ki-67 expression is also associated with a worse patient prognosis, such that Ki-67-high patients who do not respond to NAC generally have poor outcomes. Ki-67 is an indicator of the frequency of proliferative tumor cells such that higher values are consistent with more rapid tumor cell proliferation. Of the patients in this study, 31 presented with Ki-67 ≤ 14%, of whom 7 were inaccurately evaluated via BSGI, whereas all 31 were accurately evaluated via MRI. Ki-67 ≤ 14% may be linked to the overall sensitivity of the BSGI imaging modality. The proportion of residual tumors with cellularity can also impact the sensitivity of BSGI as a means of detecting residual tumor, with factors such as tumor size, cellularity, blood supply, and viability all having the potential to yield false-negative results upon scintimammography [18]. Of these seven false-negative cases in this study, five exhibited residual tumors with cellularity ≤ 10%. Consistent with this, an invasive ductal carcinoma with ER (−), PR (−), HER2 (−), and Ki-67 70% (+) status following NAC reached pCR with no evidence of residual tumor and was correctly diagnosed via BSGI (Figure 1).

Breast tumor ^99m^Tc-sestamibi uptake is primarily associated with tumor cell proliferation, angiogenesis, apoptotic gene expression, and P-glycoprotein levels. Anti-apoptotic protein levels are also negatively correlated with tumor-to-background ratio in ^99m^Tc-sestamibi-positive malignant lesions. High Bcl-2 levels are detected in an estimated 32–86% of breast carcinomas, and the overexpression of Bcl-2 can interfere with ^99m^Tc-sestamibi uptake. A delayed uptake ratio is also negatively correlated with P-glycoprotein levels prior to treatment. The ^99m^Tc-sestamibi retention index is closely associated with sensitivity to anthracyclines, indicating that double-phase scintimammography can predict breast cancer patient chemosensitivity [23,24,25,26]. We observed significant differences between the T/N ratio values in pCR patients before and after NAC treatment, whereas no such difference was observed in non-pCR patients. This suggests that changes in the T/N ratio can be used to gauge the curative efficacy of NAC in breast cancer patients to some extent. As a non-specific tumor imaging modality, BSGI also can yield false-positive results. Sun [27] et al. reported that fibrocystic changes, fibroadenoma, and benign breast tissue were the most common false-positive lesions detected by BSGI.

In this present study, invasive ductal carcinomas with HER2 positive (ER and PR negative, HER2 positive) status following NAC that reached pCR as assessed via DCIS were not correctly evaluated via BSGI and MRI. HER2 positive breast cancer usually has higher histological grade, more recurrence, and poor prognosis [28,29]. There is no evidence that the prognosis of non mass enhancement breast cancer is worse than that of mass enhancement. Gweon [30] et al. reported that HER2 positive breast cancer is more likely to be accompanied by malignant non mass enhancement lesions. Some studies have shown that non mass enhancement and multifocal or multicentric tumors are more common in HER2 positive subtypes [31,32]. This maybe take some challenges to accurately evaluate via MRI and BSGI.

Other imaging approaches such as positron emission tomography (PET) can be used alone or in combination with CT to evaluate primary breast lesion cross-sections as a functional imaging modality. However, both PET and PET/CT are unable to reliably detect lesions < 1 cm in diameter or to differentiate between benign and malignant lesions, and both are associated with high false-positive rates. PET/CT is also reported to be less sensitive as a means of detective slow-growing or low-grade tumors [33,34,35]. Recent meta-analyses have suggested that PET/CT is, however, highly specific as a predictor of breast cancer patient pathological response following NAC treatment [36,37]. While PET is both highly sensitive and specific, it can accurately predict residual disease in just 75% of cases [38], potentially owing to its low spatial resolution.

## 5. Limitations

There are a number of limitations to the present study. For one, this was a retrospective analysis of NAC treatment in breast cancer patients, and the cycles of individual patients were thus not fully controlled. Second, this was a single-center study with a small sample size, limiting our ability to evaluate the relative sensitivity and specificity of BSGI as a means of diagnosing breast cancer. Lastly, in the present analyses, L/N ratios and visual analyses were used as a means of assessing the diagnostic utility of BSGI, but these analytical approaches may have constrained our interpretations regarding the value of BSGI in this pathogenic context.

## 6. Conclusions

In conclusion, BSGI exhibits comparable diagnostic efficacy to that of MRI as a means of detecting residual tumors following neoadjuvant chemotherapy in breast cancer patients and may be an effective supplement to MRI.

## Figures and Tables

**Figure 1 diagnostics-11-01846-f001:**
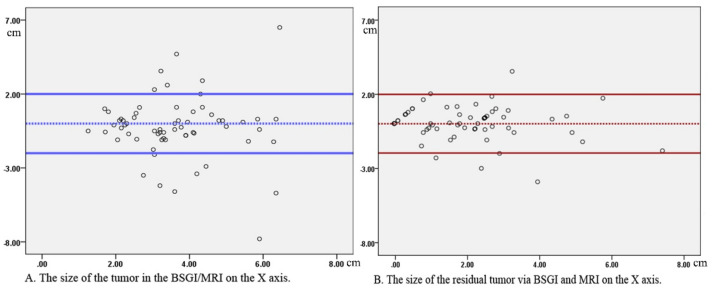
Bland–Altman plots compare maximal residual diameter between BSGI and MRI exhibited following NAC. (**A**) Before NAC. Dashed line indicates the mean difference between two methods, solid lines indicate the limits of the agreements (1.96 standard deviations of the mean difference). (**B**) After NAC. Dashed line indicates the mean difference between two methods, solid lines indicate the limits of the agreements (1.96 standard deviations of the mean difference).

**Figure 2 diagnostics-11-01846-f002:**
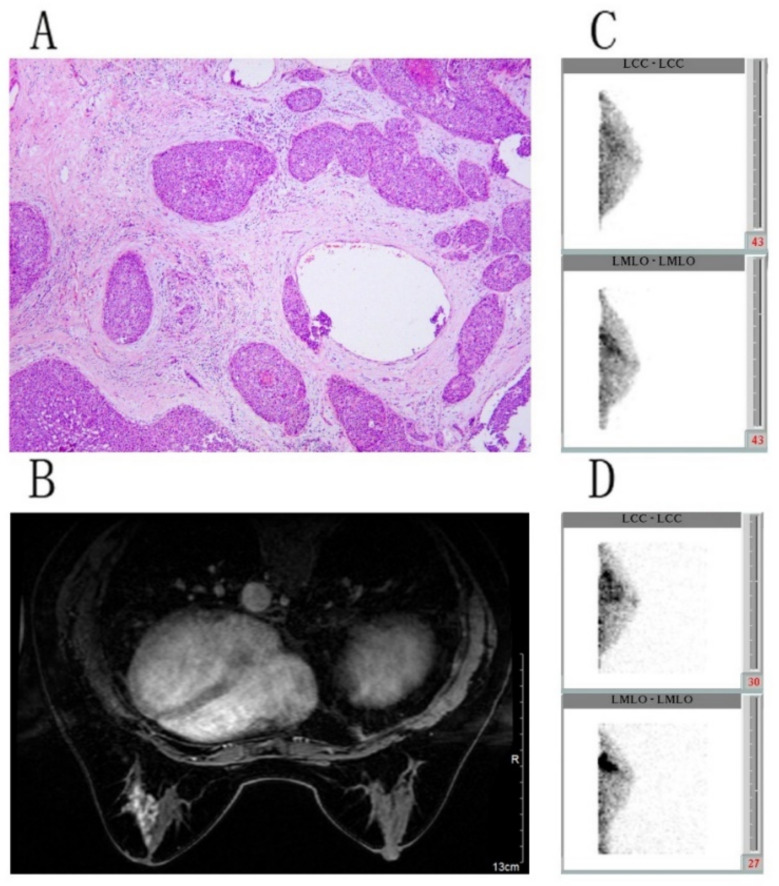
Breast scans from a 53-year-old woman. (**A**) Pathologic findings revealing DCIS (×100), necrotic tumor cells, and interstitial fibrosis (Miller-Payne grade 5, pCR). (**B**) MRI scans revealed non-mass enhancement in the left breast; efficacy evaluation was PD. (**C**) BSGI revealed clear evidence of abnormal radiotracer uptake in the left breast (maximum T/N = 1.67). (**D**) BSGI revealed a significant increase in abnormal ^99m^Tc-sestamibi foci in the left breast, consistent with metabolic disease progression (maximum T/N = 2.41; PD).

**Table 1 diagnostics-11-01846-t001:** Patient population characteristics.

Variable	Value
Age	52.8 (range 25–74)
Side of lesion	
Left breast	39 (53.4)
Right breast	34 (46.6)
Pathologic type	
IDC	68 (93.2)
Invasive lobular carcinoma	2 (2.7)
Others ^a^	3 (4.1)
NAC regimen	
TEC	25 (34.2)
EC	24 (32.9)
TCH	11 (15.1)
EC/TH	8 (11.0)
TH	3 (4.1)
AC T	2 (2.7)
Positive node status	18 (24.7)
Miller-payne classification	
G 1	11 (15.1)
G 2	13 (17.8)
G 3	19 (26.0)
G 4	13 (17.8)
G 5	17 (23.3)
Residual tumor size (cm)	1.85 ± 1.88
Estrogen receptor	
Positive	37 (50.7)
Negative	36 (49.3)
Progesterone receptor	
Positive	29 (39.7)
Negative	44 (60.3)
HER2/neu	
Positive	40 (54.8)
Negative	33 (45.2)
Ki-67	
>14%	42 (57.5)
≤14%	31 (42.5)
Molecular subtype	
Luminal A	12 (16.4)
Luminal B	33 (45.2)
Her-2 (+)	22 (30.1)
Triple-negative	6 (8.3)
Total	73 (100)

IDC invasive ductal carcinoma; EC epirubicin plus cyclophosphamide, TEC docetaxel, epirubicin, plus cyclophosphamide, TH docetaxel plus trastuzumab, TCH docetaxel, carboplatin, plus trastuzumab, AC T doxorubicin, cyclophosphamide plus docetaxel. SD standard deviation. ^a^, tubular carcinoma (*n* = 1), intraductal micropapillary carcinoma (*n* = 1), mucinous carcinoma (*n* = 1).

**Table 2 diagnostics-11-01846-t002:** The tumor to normal tissue (T/N) ratio values for the pCR and non-pCR groups.

Group	Before NAC	Post NAC	t Value	*p*
pCR	3.25 ± 0.96	1.49 ± 0.30	3.997	0.044
npCR	3.26 ± 1.18	2.61 ± 1.79	2.153	0.543
t value	0.053	1.465		
*P*	0.501	0.180		

**Table 3 diagnostics-11-01846-t003:** The maximal residual diameter between BSGI and MRI exhibited following NAC.

	Modality	Mean ± SD	Median	Range	t	*p*
Before NAC	MRI BSGI	3.49 ± 1.63 3.77 ± 1.73	3.19 3.61	1.03–9.67 1.21–9.83	−1.049	0.298
After NAC	MRI BSGI	1.92 ± 1.59 1.90 ± 1.73	2.06 2.30	0.22–6.58 0.59–8.33	0.126	0.900

SD, standard deviation.

**Table 4 diagnostics-11-01846-t004:** Statistics regarding the consistency between BSGI, breast MRI, and pathological measurements of residual tumor following NAC treatment.

	N	Sensitivity (%)		
		BSGI	MRI	*p*
Residual cellularity				
≤10%	21	16 (76.2) ^a^	17 (80.9) ^b^	0.707
>10%	35	33 (94.3)	34 (97.1)	0.555
Residual tumor size				
≤15 mm	25	19 (79.2) ^c^	20 (80.0) ^cd^	0.733
>15 mm	31	30 (96.8)	31 (100.0)	0.313
Molecular subtype				
Luminal A	11	10 (90.9)	9 (81.8)	0.534
Luminal B	27	23 (85.2)	26 (96.3)	0.159
Her-2 (+)	14	13 (92.9)	12 (85.7)	0.541
Triple-negative	4	3 (75.0)	4 (100)	0.285
ER Express				
Positive	37	31 (83.8)	33 (89.2)	0.496
Negative	19	18 (94.7)	17 (89.5)	0.547
PR Express				
Positive	25	22 (88.0)	24 (96.0)	0.297
Negative	31	27 (87.1)	27 (87.1)	1
HER-2 Express				
Positive	41	35 (85.4)	37 (90.2)	0.5
Negative	15	14 (93.3)	14 (93.3)	1
Ki-67				
>14%	42	10 (23.8)	13 (31.0)	0.463
≤14%	31	7 (16.7) ^d^	0 (0)	0.005
Invasiveness				
Invasive residual	49	42 (85.7)	45 (91.8)	0.337
In situ residual	7	7 (100)	6 (85.7)	0.299
Total	56	49 (87.5)	51 (91.1)	0.541

^a^ ≤10% Group to >10% Group, *p* = 0.047; ^b^ ≤ 15 mm Group to >15 mm Group, *p* = 0.019; ^c^ ≤15 mm Group to >15 mm Group, *p* = 0.009; ^d^ >14% Group to ≤14% Group, *p* = 0.902.

**Table 5 diagnostics-11-01846-t005:** The evaluation of BSGI, MRI, and pathological measurements of residual tumor following NAC treatment.

Case No.	Age	Subtype	Chemotherapy Regimens	M-P Grade	Residual Size (cm)	MRI (cm)	MRI Evaluation	BSGI (cm)	T/N	BSGI Evaluation
1	58	HER-2	EC	1	1.0	2.6	PD	2.4	1.97	PR
2	44	Luminal A	TEC	2	1.0	1.7	PD	1.8	1.91	PR
3	51	Luminal B	TEC	3	1.0	1	PR	0	1	CR
4	64	Luminal B	TCH	3	0.7	1	PR	0	1	CR
5	52	Luminal B	TEC	3	0.5	0.76	PR	0	1	CR
6	57	Luminal A	TEC	3	0.3	1	SD	0	1	CR
7	67	Luminal A	TCH	3	1.2	2	SD	0	1	CR
8	46	Luminal A	TEC	3	1.6	2.4	SD	0	1	CR
9	65	Luminal A	TCH	3	0.4	0.9	SD	0	1	CR
10	45	Luminal B	TEC	3	4.5	0.5	PR	0	1	CR
11	74	Luminal B	EC	3	2.0	3.1	SD	2.1	2.76	PR
12	49	HER-2	TCH	4	1.0	0	CR	0	1	CR
13	49	HER-2	TCH	4	0.5	0	CR	0	1	CR
14	39	Luminal B	TEC	4	0.5	0	CR	0	1	CR
15	59	HER-2	TEC	4	0.1	1.6	PR	0	1	CR
16	59	HER-2	TEC	4	0.1	0.6	PR	0	1	CR
17	48	HER-2	TCH	4	0.1	2.2	PR	0	1	CR
18	53	HER-2	TCH	5	2	non-mass enhancement	PD	4.5	2.41	PD
19	70	Luminal A	TEC	5	0.1	1.9	PR	2.3	1.38	PR
20	60	HER-2	EC/ TH	5	a small focus of residual DCIS	2.0	PR	1.3	1.26	PR
21	54	HER-2	TCH	5	0	2.3	PR	1.0	1.3	PR
22	54	HER-2	EC	5	0	0.6	PR	0.9	2.36	SD

## Data Availability

The data involved in the current study are available upon request. Anyone who is interested in the information should contact hbliu@zju.edu.cn.

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
