# Peer review of "Comparison of BSGI and MRI as Approaches to Evaluating Residual Tumor Status after Neoadjuvant Chemotherapy in Chinese Women with Breast Cancer"

_diagnostics, 2021, doi:10.3390/diagnostics11101846_

Round 1
Reviewer 1 Report
In Bland-Altman plots, notations for the X and Y axes are required. It seems better to enter the size of the pathological residual tumor on one axis and the size of the tumor in BSGI/MRI on the other axis.
Author Response
Responds to the reviewer’s comments:
Response to comment: In Bland-Altman plots, notations for the X and Y axes are required. It seems better to enter the size of the pathological residual tumor on one axis and the size of the tumor in BSGI/MRI on the other axis.
Response: We have revised this part according to the Reviewer’s suggestion.

Reviewer 2 Report
This manuscript improved a lot compared to the first edition. I suggest to accept this manuscript after following minor revisions:
- Figure 1: please make y-axis of these two plots on the same scale
- Figure 2&3 were not mentioned in text. Better place them in Results, not Discussion
- Last sentence on page 9 (line 292) is not clear. Please rephrase it.
Author Response
Responds to the reviewer’s comments:
- Response to comment: Figure 1: please make y-axis of these two plots on the same scale
Response: We have revised this part according to the Reviewer’s suggestion.
- Response to comment: Figure 2&3 were not mentioned in text. Better place them in Results, not Discussion
Response: We have added the content according to the Reviewer’s suggestion.
- Response to comment: Last sentence on page 9 (line 292) is not clear. Please rephrase it.
Response: We have revised this part according to the Reviewer’s suggestion.
This manuscript is a resubmission of an earlier submission. The following is a list of the peer review reports and author responses from that submission.
Round 1
Reviewer 1 Report
This is a retrospective study of breast-specific gamma imaging (BSGI) in the evaluation of residual tumor after neoadjuvant chemotherapy (NAC) and it was compared with the performance of MRI. BSGI is a nuclear-based imaging for the breast which uses a radiation tracer. Both BSGI and MRI are physiologic approaches to image breast cancer. In this paper, a total of 390 women who underwent BSGI at a hospital in China were included. Among these women, 73 were included in the analysis. The authors reported the sensitivity value of 76.8% for BSGI, compared to 69.6% for MRI. Specificity values were also reported as 70.6% versus 58.8%, respectively. The conclusion was that BSGI can be used as a supplemental tool to MRI in the evaluation of tumor residual after NAC.
In my opinion, the motivation of this study was not compelling. After reading the Background, it is not clear to me why BSGI is studied since MRI is “generally considered to be the ideal imaging modality ….”. The arguments of “even MRI-based analyses may significantly misjudge the amount of residual tumor in treated patients” and “BSGI is a high-resolution radioimaging strategy that enables breast tissue visualization using a gamma camera with a limited field-of-view” are not strong reasons to conduct the study.
Please give more details about MRI in section 2.2. And the method of how tumor residual in BSGI was evaluated was not clear. It seems like these “two nuclear medicine specialists” also used baseline mammography, ultrasound, and MRI but not clear how. For fair comparison, should BSGI and MRI both evaluated independently?
Table 4 in Results was not interpreted. It is difficult to understand. No figures should be included in Discussion.
The following comments are for minor issues in the paper:
- Add numbers next to the sensitivity and specificity in the Abstract
- Add reference after sentence on Page 1 line 29-32
- Add reference after sentence on Page 1 line 36-38
- Add reference after sentence on Page 9 line 269-270
- In Section 2.4, there is a mixture of pathological evaluation and imaging evaluation of tumor response to treatment. It’s difficult to understand
- Line 119 on Page 3: P<0.05 should be P=0.05
- Line 269 on Page 9 “Lastly,…” is not a limitation
Reviewer 2 Report
There are not many studies on the accuracy of BSGI after NAC, so it is considered a good study.
Please make some corrections.
1. Please add a reference to the following sentence.
‘NAC can significantly increase rates of overall and disease-free survival (OS and DFS, respectively) similarly to postoperative chemotherapy, while increasing rates of breast-conserving surgery in those with operable locally advanced disease’
‘NAC treatment can also decrease the extent of resection in cases where tumors are over 2 cm in size’
2. 'BSGI images' Breast-specific gamma imaging: Words overlap.
3. Table 4 contains data for only a subset of patients. What will be the selection criteria?
4.It is recommended to add the 'Bland-Altman plot' to compare the accuracy according to the size of the tumor in the two imaging tests.
5.Abstract:'BSGI and MRI were associated with respective 76.8% and 69.6% sensitivity values as a means of detecting residual tumors following NAC, while both of these approaches exhibited comparable specificity in this diagnostic context(58.8% vs 70.6%, P=0.473)'
Results:'BSGI and breast MRI scans exhibited respective sensitivity values of 69.60% (39/56) and 76.8% (43/56) for the detection of residual tumor, while corresponding specificity values were 70.6% (12/17) and 58.8% (10/17)'
-The content of the abstract and the content of the body do not match.